# Improvement of Acoustic Models Fused with Lip Visual Information for Low-Resource Speech

**DOI:** 10.3390/s23042071

**Published:** 2023-02-12

**Authors:** Chongchong Yu, Jiaqi Yu, Zhaopeng Qian, Yuchen Tan

**Affiliations:** School of Artificial Intelligence, Beijing Technology and Business University, Beijing 100048, China

**Keywords:** audiovisual speech recognition, low-resource language, automatic speech recognition, lipreading

## Abstract

Endangered language generally has low-resource characteristics, as an immaterial cultural resource that cannot be renewed. Automatic speech recognition (ASR) is an effective means to protect this language. However, for low-resource language, native speakers are few and labeled corpora are insufficient. ASR, thus, suffers deficiencies including high speaker dependence and over fitting, which greatly harms the accuracy of recognition. To tackle the deficiencies, the paper puts forward an approach of audiovisual speech recognition (AVSR) based on LSTM-Transformer. The approach introduces visual modality information including lip movements to reduce the dependence of acoustic models on speakers and the quantity of data. Specifically, the new approach, through the fusion of audio and visual information, enhances the expression of speakers’ feature space, thus achieving the speaker adaptation that is difficult in a single modality. The approach also includes experiments on speaker dependence and evaluates to what extent audiovisual fusion is dependent on speakers. Experimental results show that the CER of AVSR is 16.9% lower than those of traditional models (optimal performance scenario), and 11.8% lower than that for lip reading. The accuracy for recognizing phonemes, especially finals, improves substantially. For recognizing initials, the accuracy improves for affricates and fricatives where the lip movements are obvious and deteriorates for stops where the lip movements are not obvious. In AVSR, the generalization onto different speakers is also better than in a single modality and the CER can drop by as much as 17.2%. Therefore, AVSR is of great significance in studying the protection and preservation of endangered languages through AI.

## 1. Introduction

Endangered languages, the ones verging on extinction, are cultures’ important carrier and component. As extinction of languages incurs irreparable damages on the world’s language resources, protecting endangered languages is, thus, critical. Such an attempt is mainly achieved through collecting and labeling audio and video materials of native speakers. However, without sufficient native speakers, non-native speakers, despite their best effort, cannot accurately label the materials collected, due to the lack of expertise, protocols, and technology. The state-of-art ASR opens up a new path for automatic labeling and protection of endangered languages. Studies show that using ASR instead of manual labeling can greatly help linguists to protect endangered languages. [1,2,3]. This paper focuses on Tujia language as the case in point to explore the modeling for more accurate ASR of low-resource language.

ASR converts speech to text using computer algorithms and, thus, replaces manual labeling. ASR has been widely applied to majority languages. ASR’s acoustic models are critical, especially for endangered unwritten languages. The Traditional Hidden Markov Model (HMM) [4], Gaussian Mixture Model (GMM) [5], etc., are based on state transitions. Sequence processing models such as Long Short-Term Memory (LSTM) [6], Connectionist Temporal Classification (CTC) [7], and Transformer [8] rely on a typical end-to-end approach. At the early stage, Tachbelie et al. [9] used HMM to achieve acoustic modeling of Amharic (an endangered language) and used syllables as the modeling unit. Amharic speech was automatically labeled as syllables and recorded accordingly. However, HMM relies on frame-by-frame training whose output is single-frame frequencies. Therefore, the model was deficient in utilizing the contextual semantic relations of speech sequences. To make up for this deficiency, researchers chose sequence-based, end-to-end deep learning in their modeling. For instance, Inaguma et al. [10] resorted to BLSTM-CTC to recognize languages with insufficient data. They also incorporated transfer learning, drawing upon knowledge learnt from corpora of majority languages for more accurate recognition. The performance of such an approach is even better than the BLSTM-HMM hybrid model. However, transfer learning is not applicable to the recognition of two languages whose acoustic space differs substantially. Transfer learning aims to draw upon prior knowledge when insufficient data result in insufficient learning. With a great difference between two languages, the learning of the target language would be hampered. As Tujia language and Chinese thchs30 [11] are both Sino-Tibetan languages, we use Chinese corpora as an extension of the Tujia language and construct a cross-language corpus. Then, we use transfer learning to establish the model of the cross-language end-to-end Tujia language recognition system [2]. Experimental results show that such an approach can improve the accuracy of recognizing Tujia language, but to a limited extent. The overall accuracy is still not ideal. Meanwhile, field study is often necessary to acquire raw data for most endangered languages, in which case noise would become an element beyond control. Acoustic models would be disturbed, resulting in deteriorating performance. Speech enhancement, currently the major approach to reduce noise’s influence, is applied mostly to speech recognition of majority languages [12,13,14,15]. Yu et al. [16], applying speech enhancement to endangered languages, put forward an end-to-end speech enhancement model based on an improved deep convolutional generative adversarial network (DCGAN) to improve speech quality for Tujia language. For enhanced speeches, Yu et al. undertook the perceptual evaluation of speech quality and calculated the mean opinion score of the listening quality objective. However, they did not apply speech enhancement during recognition. The performance for such an application still needs to be verified.

In addition, protecting endangered languages requires sustained effort. Therefore, models’ generalization onto new speakers, or speaker adaptation, is necessary. Speaker-adapted recognition means that models’ parameters can reflect features of multiple speakers. Research on speaker adaptation can be divided into two categories. The first is speaker adaptation based on feature domain. During the training phase, Ochiai et al. [17] retained the acoustic model’s parameters while changing speakers’ features to improve the adaptability of models onto features. The parameters are re-evaluated when adapting to target speakers. However, overfitting may occur when the data are insufficient for adaptation training, leading to dissatisfactory results of adaptation. The second is speaker adaptation based on model domain. Models’ parameters are updated to match the adaptive data of the target speaker. Regularization [18,19], Kullback–Leibler divergence [20,21], and adversarial multitask learning [22] are applied to avoid overfitting. However, the second kind of speaker adaptation requires a large number of speakers as excessive divergence of updated parameters from the original models should be avoided during speaker adaptation. Such a large number of speakers are difficult to gather for majority languages and impossible to achieve for endangered languages with scarce native speakers. To sum up, traditional approaches of speaker adaptation require a large amount of data of speakers and are, thus, not applicable to endangered languages with scarce data or speakers. 

The McGurk effect [23] shows that human beings pay attention to their interlocutors’ lip movement for better understanding. For a long time, ASR did not require such visual perception. Current recognition approaches mainly rely on acoustic models. However, these approaches’ accuracy is reaching the limit, with no substantial improvement in sight [24]. As a result, multi-modal approaches are spreading. Audiovisual fusion speech recognition (AVSR), put forward based on the bimodal speech perception mechanism [25], incorporates the visual modality in addition to the audio mode to jointly complete the transcription from speech to text. In AVSR, the visual modality offers complementary information on the basis of the audio modality. When errors occur in one modality, complementary information can be drawn upon to improve accuracy. The overall uncertainty is, thus, lower than that of single-modality recognition. As it adds more information, AVSR becomes a feasible option to address the insufficient training and low accuracy of single-modality recognition and make up for the scarce data and labeling of endangered languages. AVSR reduces the reliance on the audio modality and the introduced visual modality aims to extract relevant features by capturing the deformation of the speaker’s lips when speaking to mine the information of the discourse content. As a result, AVSR still functions well in noisy environments, which tackles the deteriorating performance when noises disturb the recognition of endangered languages [26,27]. Meanwhile, people differ both acoustically and visually even pronouncing the same phoneme. The combination of information in both visual and audio modalities can reduce the speaker-specific influence on models’ accuracy, leading to a model widely applicable to the recognition of endangered languages. Based on the above analyses, the paper proposes the use of AVSR to tackle difficulties such as scarce data and labeling, noises, and insufficient speakers in the ASR of low-resource language. The paper’s contributions are as follows.

(1)Focusing on low-resource language, an endangered language, the paper proposes AVSR to tackle the difficulties due to scarce speech data and labeling, noises, and insufficient speakers (in the case of single-modality recognition). In addition to single modeling, AVSR draws upon lip movement for complementary information to achieve multi-modal learning.(2)The paper proposes an end-to-end model based on LSTM-Transformer. In training, the model can not only learn the contextual relation of the sequence signal, but also learn the temporal and spatial correlation between them when the fusion information is diverse and complex.(3)Speaker-related experiments are designed to verify that the proposed method can solve the speaker dependence problem by comparing the recognition accuracy when the training set contains the speakers in the test set and the training set does not contain the speakers in the test set.

## 2. Audiovisual Fusion Model for Tujia Speech

The paper applies audiovisual fusion to recognize the low-resource language to improve accuracy. The proposed AVSR is shown in Figure 1. The visual image sequence is processed by a 3D-2D convolution, while the audio features are the spectrograms obtained by applying the Short-Time Fourier Transform (STFT) to the audio signal. The audio and video features serve as the model’s input. Each modality is encoded by a separate LSTM encoder and the concatenated audio and video encodings are then decoded by the Transformer decoder. In this way, multi-modal fusion based on audiovisual encoding is, thus, achieved. At the final stage, CTC is applied for the model’s predicted characters.

### 2.1. Feature Extraction

In networks of 2D convolution, convolution is applied to static images and features are extracted only in the spatial dimension. However, we gather speakers’ video as the corpus source for low-resource language, which, thus, becomes the basis for analysis. As the image sequence is consecutive and relevant, obtaining the relationships between the images is key to video data analysis. To summarize, we resort to 3D convolution [28] to extract video features in both the spatial and temporal dimension. In the final stage, 3D feature maps are passed down through 2D convolution for down sampling and are, thus, downgraded to the same dimension as audio features.

We apply STFT for the transformation from the time domain to frequency domain to extract features of speech signals. With consecutive temporal signals, we multiply signals not yet transformed by a window function that is not zero-valued only for a short interval and then apply the one-dimensional Fourier transform. Supposing that signals are time-invariant within the time window, we slide the window along the time axis and put the results of the one-dimensional Fourier transform in sequence. In this way, we obtain the 2D expression of the untransformed signals, and we use a hamming window here. As it can eliminate high-frequency interference and energy leakage, the hamming window is applicable to aperiodic continuous-speech signals.

### 2.2. Audiovisual Fusion Block

The key to AVSR is how the fusion of audio and video modalities is achieved. Early-stage approaches reviewed by Katsaggelos et al. [29] can be categorized as decision fusion and feature fusion. In decision fusion, video and audio modalities are assumed to be completely independent. The independent recognition of two feature streams is combined in the final decision-making [30,31,32,33]. However, the correlation between video and audio modalities is neglected. In feature fusion, audio and video features are combined at the early stage as new audiovisual features to be learnt [34,35,36,37]. Although the correlation between audio and video modalities is drawn upon, their mutual independence is overlooked. Meanwhile, the end-to-end AVSR tends to perform audiovisual feature fusion on higher-level representations of each modality after the encoding process [38,39,40]. Compared with feature and decision fusion, this approach maintains the asynchronous relationship between audiovisual data and, modeling with certain independence, ensures the reliability of features in each modality. It is able to provide synchronous information of audio and video modalities and does not require forced alignment of the two modalities [41]. Therefore, we choose multi-modal fusion based on audiovisual encoding for the speech recognition of low-resource language.

The low-resource language that we study is subject to phonological changes where the pronunciation of a syllable is changed by the previous syllable. Such changes mean that it enjoys diverse and complex features. Noises may also lead to slightly different features even for the same pronunciation. Therefore, the requirement is high for models to learn temporal information within the context. LSTM, a special Time Recurrent Neural Network, is suitable for the learning of time sequence. The memory cell retains each step’s information for the dynamic alignment with the next step. In this way, LSTM retains information of a local sequence. Meanwhile, the forget gate selectively forgets unimportant information and memorizes important information. The input at every step, thus, takes into consideration the previous input. At a specific time t, a LSTM neuron is calculated as in Formulae (1)–(6).
(1)$$i_{t}=\sigma \left(W_{ix}x_{t}+W_{ih}h_{t-1}+W_{ic}c_{t-1}+b_{i}\right)$$
(2)$$f_{t}=\sigma \left(W_{fx}x_{t}+W_{fh}h_{t-1}+b_{f}\right)$$
(3)$$c_{t}=f_{t}c_{t-1}+i_{t}\phi \left(W_{cx}x_{t}+W_{ch}h_{t-1}+b_{c}\right)$$
(4)$$u_{t}=\sigma \left(W_{ux}x_{t}+W_{uh}h_{t-1}+W_{uc}c_{t}+b_{u}\right)$$
(5)$$h_{t}=u_{t}\phi \left(c_{t}\right)$$
(6)$$y_{t}=LSTM\left(t\right)=W_{yh}h_{t}+b_{y}$$
where, at a specific time *t*, $i_{t}$, $f_{t}$, $c_{t}$, $u_{t}$, $h_{t}$, $x_{t}$, and $y_{t}$ stand for the input gate, forget gate, memory cell, output gate, and hidden state, as well as input and output, respectively. W is the weight matrix linking different gates with b being the corresponding bias vector. σ is a sigmoid function and ϕ is a tanh function.

Using LSTM to encode input audio and video features, we obtain $x_{A}=\left\{x_{A_{1}},x_{A_{2}},x_{A_{3}},\ldots ,x_{A_{N}}\right\}$, a phonological sentence of variable length, and the corresponding video stream $x_{V}=\left\{x_{V_{1}},x_{V_{2}},x_{V_{3}},\ldots ,x_{V_{N}}\right\}$. At each step, we obtain the audio feature $f_{i}^{a}$ and the video feature $f_{i}^{v}$. After Formulae (7) to (9), we obtain the codes after audiovisual fusion.
(7)$$h_{i}^{a},o_{i}^{a}=LSTM\left(f_{i}^{a}\right)$$
(8)$$h_{i}^{v},o_{i}^{v}=LSTM\left(f_{i}^{v}\right)$$
(9)$$o_{i}^{av}=concat\left\{\left(h_{i}^{a},o_{i}^{a}\right),\left(h_{i}^{v},o_{i}^{v}\right)\right\}$$
where $o_{i}^{a}$ and $o_{i}^{v}$ stand, respectively, for the audio and video coding after LSTM. $o_{i}^{av}$ is the coding after fusion at each step. $h_{i}^{a}$ and $h_{i}^{v}$ are the short-term sequence information retained.

The connected audio and video codes incorporate the sequential information learnt in LSTM. However, the information just includes local contexts. As sentence inversion is an important feature of low-resource language that we study, syntactic structures also need to be learnt. Therefore, further learning is necessary for the integrated information for better correlation. Transformer’s multi-head attention mechanism helps models to learn information in different subspaces. After fusion of input information, each information can perform attention operation with other fusion information to learn the interdependence between corresponding audio and video information. Such a structure also helps the multi-head mechanism to grasp the overall situation. Meanwhile, Transformer makes parallel computing possible when models learn sequential information. Every module can compute simultaneously, substantially reducing the time and resource cost for processing fusion encoding, covering a large amount of information. Therefore, we use Transformer to decode and LSTM to learn sequential information of the context and remove positional encoding.

Specifically, integrated audio and video codes $o^{av}=\left\{o_{1}^{av},o_{2}^{av},o_{3}^{av},\ldots ,o_{N}^{av}\right\}$ are put into Transformer. Through multi-head attention, as in Formula (13), weights are assigned to important information. Noises and redundancies in the input are ignored for comprehensive decision-making.
(10)$$c_{i}=MultiHead^{av}\left(o^{av}\right)$$
where $c_{i}$ stands for output context vectors.

### 2.3. TM-CTC Decoder

We use TM-CTC to decode the International Phonetic Alphabet (IPA) characters of low-resource language. Compared with other decoders, CTC no longer needs frame-level segmentation and alignment for training data and does not need modeling units to output target transcript sequences [42]. Therefore, CTC is suited for decoding long speech sequences. Meanwhile, CTC adds an extra blank label to the target label set and uses the blank label to show the probability of not producing any labels within a specific stride length.

Specifically, for an input sequence $y=\left\{y_{1},y_{2},\ldots ,y_{N}\right\}$ of length *T* and an output sequence $\;y^{\prime }=\left\{y_{1}^{\prime },y_{2}^{\prime },\ldots ,y_{N}^{\prime }\right\}$ of length *N*, we define a target label set Ω ($y_{n}^{\prime }\in \Omega$) and an extended CTC target label set $\underset{\Omega }{¯}=\Omega \cup \left\{-\right\}$. Through CTC, y is projected to path $\pi =\left\{\pi_{1},\ldots ,\pi_{T}\right\}$ of the same length T ($\pi_{t}\in \underset{\Omega }{¯}$). The conditional probability for an arbitrary path π is defined as in Formula (11).
(11)$$P\left(\pi |y\right)=\prod_{t=1}^{T}P\left(\pi_{t}|y\right)$$

The set of path π of length *T* is recorded as $B^{-1}\left(y^{\prime }\right)$ and CTC needs to achieve many-to-one and long-to-short projections. Multiple paths are combined into shorter label sequences as consecutive and identical labels in every path are combined as one and blank labels are deleted. Meanwhile, the probability for the target label sequence needs to be calculated, as in Formula (12).
(12)$$P\left(y^{\prime }|y\right)=\underset{\pi \in B^{-1}\left(y^{\prime }\right)}{\sum }P\left(\pi |y\right)$$

Finally, a greedy search algorithm is used to output predictions with CTC.

## 3. Experiments

### 3.1. Dataset

For low-resource language datasets, we collect the following materials: 52 oral materials of 9 h, 54 min, and 39 s; only 10 of the 52 corpus have recorded videos of 1 h, 20 min, and 45 s, acquired by four speakers. The oral materials can be categorized as lexis and grammars, oral culture, narratives, and dialogues.

We set the parameters of recorded audios to two channels and a 16 kHz sampling rate. To reduce difficulties in dual-channel recognition and maintain data consistency, we extract the left channel information in batches from the audio files. We segment long speeches of an individual speaker at intervals into short speeches. We convert the contents annotated per sentence with ELAN into files of Praat TextGrid format, incorporating the starting and ending times of each sentence. With Praat and script, we segment long speeches into short ones and eliminate blanks between sentences. In all, we segment long speeches into 10,348 short sentences, used for the single-modal experiments in Section 4.1.

For text files, we export the IPA layer of each oral material with ELAN to obtain a text file and then segment per line the file into multiple files. We then obtain the IPA-labeled files for every sentence. Original manual annotations include punctuations. To eliminate such an influence on the recognition rate, we filter out all punctuations in the text files and eliminate digital tones in superscript. Finally, we obtain 10348 IPA-labeled text files corresponding to the audio ones and all text files are in UTF-8 format.

The frame rate of recorded videos is 25 frames per second (648 in height and 1168 in width). As the environment for video-recording is complex, we crop the background and obtain 160 × 160 videos centering on speakers’ heads. Based on the starting and ending times exported from ELAN, we use python to segment videos per sentence and align the segmented videos with the audio files. Because not all the audio in our corpus has corresponding videos, we finally obtain 2105 short videos, used for multi-modal experiments in Section 4.2 and Section 4.3 with corresponding audios.

Datasets with videos in the paper are shown in Table 1.

### 3.2. Setup Conditions for Initials and Finals

As Tujia language does not have a written form, IPA is used to represent phonemes and organize the pronunciation into 21 initials, as shown in Table 2, and 25 finals, including 6 unitary finals, 11 complex vowels, and 8 nasalized vowels and finals, as shown in Table 3.

### 3.3. Evaluation Metrics

The evaluation indexes are the Character Error Rate (CER) per IPA character. The calculation is shown as in Formula (13).
(13)$$\mathrm{CER}=\frac{LevenshteinDistance\left(predict,label\right)}{len\left(label\right)}$$
where the Levenshtein Distance predicts the edit distance between the predicted and actual character sequence. The calculation of the edit distance is shown as in Formula (14).
(14)LevenshteinDistance=S+D+I
where *S* shows the number of characters to be replaced, *D* shows that to be deleted, and *I* shows that to be inserted. The lower the CER is, the better the recognition results are.

### 3.4. Experimental Setup

#### 3.4.1. Visual Features

We extract visual features in the following phases, face detection, ROI extraction, post-processing, and feature extraction. We put in 160 × 160 videos and resort to CNN and single-shot MultiBox Detector (SSD) for face detection [43]. As facial movements are frequent and complex, SSD helps to observe faces from multiple perspectives, which avoids the loss of human faces. We then label the 112 × 112 area around the lips and undertake gray processing for the cropped area. To extract the visual features of lip movement, we apply 3D convolution to the input image sequence, with the convolution kernel of 5 × 7 × 7, the convolution stride of 1 × 2 × 2, and a last-stage padding of 2 × 3 × 3. Next, we use 2D convolution for down sampling to obtain the 512-dimensional features of lip movement. We repeat the above-mentioned processes every 40 ms and the processes are illustrated in Figure 2.

#### 3.4.2. Acoustic Features

For acoustic features, we set the sampling rate to 16 kHz and apply the hamming window and STFT. We set the frame length to 40 ms and frame shift to 10 ms. We extract a STFT feature every 10 ms to obtain a 321-dimensional spectrogram.

#### 3.4.3. LSTM-Transformer

The LSTM-based encoding module uses 3 layers of LSTM with 512 hidden nodes at each layer. The Transformer-based decoder uses a multi-head attention mechanism that incorporates 6 self-attention mechanisms. The mechanisms’ Q, K, and V are the input feature tensors. The whole process includes 8 heads and 512 nodes at each layer and abandons the positional encoding module.

#### 3.4.4. Sequences Alignment

As the dataset of Tujia language segments speeches with sentences as units, videos and audios are, thus, not of the same length. However, the convolutional module in the experiment has a high requirement for the alignment of the matrix’s dimensions. Therefore, zero padding alignment is necessary for different audios and videos. The length of the padding is defined by the length of the dataset’s labels, with the length of the longest label marking the shortest length of padding.

For the extraction of audio and visual features, the sampling speed is 40 ms per frame for videos and 10 ms per frame for audios. The alignment of audio and visual features is, thus, achieved with 4 consecutive audio features corresponding to 1 visual feature. In this way, the length of the input sequence is reduced, which helps CTC decoding.

#### 3.4.5. Speakers Preparation

The dataset includes 4 speakers of different age and gender. Differences in speakers’ frequency and timbre lead to differences in extracted features of the same word. Differences also exist in lips’ deformation.

When different speakers say the same word, considering such differences, when designing speaker-related experiments, we allocate the training set and test set according to a 9:1 ratio. The training set covers 3 speakers and the test set covers the remaining 1 speaker.

(a) and (b) in Figure 3 are the spectrograms of the ai pronunciation of two different speakers. The difference can be clearly seen in the labeling part, and the energy of the pronunciation in (a) is significantly higher than that in (b).

(a) and (b) in Figure 4 are the lip shapes of two different speakers pronouncing ai. It can be seen that the lip shape of (a) is flat, and the lip movements are not obvious. In contrast, the lip shape is rounded in (b), and the lip movement is more obvious.

## 4. Results and Analysis

Results for evaluations are listed in Table 4, Table 5 and Table 6, and all the evaluations are calculated based on the Tujia language. The experimental results in Table 4 are obtained from only audio data, and the primary purpose is to find the modeling units (phonemes or characters) that are more suitable for the Tujia language, so all audio data are used. The data gradient in Table 5 selects 1000, 1400, and 1800 because the multi-modal corpus of this study only contains 2105 sentences (mainly limited by the video data).

### 4.1. Results of Acoustic Model

To verify the effectiveness of ASR in automatic labeling of low-resource language, we undertake experiments based on state transition and sequence modeling. In state transition modeling, we use a monophone as an acoustic unit and extend to a triphone model. As the low-resource language that we study does not have a written form, we resort to IPA in labeling. Therefore, in sequence modeling based on deep learning, we use IPA characters as the modeling unit. Experimental results are shown in Table 4.

The CER of triphone modeling is 0.18% to 2.23% lower than that of monophone modeling for the recognition of 2000, 5000, and 6000 sentences and is 3.15% and 1.13% higher for the recognition of 3000 and 4000 sentences, respectively. The result shows that when data are relatively scarce, adjacent phonemes exert an influence on the recognition accuracy, but to a limited extent.

The CER of the Transformer-CTC model is 24.02% lower than that of the HMM-based model for the recognition of 2000 sentences and is 19.32% to 25.74% lower for the recognition of 6000 sentences. The result shows that compared with other models, Transformer-CTC does not see its performance substantially improve with more data. However, for Transformer-CTC alone, it sees the CER drop by 17.8% when recognizing 2000 and 6000 sentences. This shows that more data can effectively improve Transformer-CTC’s performance.

A basic assumption of monophone modeling is that the actual pronunciation of a single phoneme is unrelated to adjacent or close phonemes. However, this assumption does not stand for languages with co-articulation, such as the Tujia language.

For the initials of the Tujia language, the stops and fricatives are only voiceless and not voiced, but voiceless initials often appear voiced in the speech flow.

The final i of the Tujia language has two phonetic variants, i and ɿ. The final is pronounced as ɿ after ts, ts^h^, s, and z and as i after other initials. For instance, tsi 21 is pronounced as tsɿ21, ts^h^i53 as ts^h^ɿ53, si53 as sɿ53, and zi55 as zɿ55. The final e does not go with k, k^h^, x, and ɣ and follows an obvious medial i when pronounced with other initials. However, it follows the medial i but a slight glottal stop in zero-initial syllables. For instance, pe35 is pronounced as pie35, me35 as mie35, and te35 as tie35. In compound words, if the initials are voiceless stops or affricates, they are influenced by the (nasal) vowels of previous syllables and become voiced stops and affricates. Such a phenomenon takes place normally in the second syllable of bi-syllabic words and does not influence aspirated voiceless stops or affricates. For instance, no53pi21 is pronounced as no53bi21 and ti55ti53 as ti55di53.

The examples show that for Tujia language, the actual pronunciation of phonemes is influenced by adjacent or similar phonemes and may change due to different positions of the phonemes. Therefore, using central, left-adjacent, and right-adjacent phonemes as the basis for modeling and recognition can improve the recognition rate of Tujia language. However, as the amount of data decreases, the model may be unstable and over-fitted, due to little data and many parameters.

Experimental results show that sequence modeling is better than state-transition modeling. Therefore, sequence modeling is better suited for the recognition of low-resource language. Meanwhile, phonological changes are common in the low-resource language used and are not compulsory. Such changes are closely related to other sentence structures and, thus, have a significant influence on frame-based phoneme modeling, making overall optimization impossible. Therefore, for speech recognition of low-resource language, sequence modeling shows better performance.

### 4.2. Results of AVSR

To explore the audiovisual fusion suited to low-resource language and verify the effectiveness of the approach put forward in this paper, we design experiments comparing audio and visual recognition models, including a single audio or visual modality, Transformer-based AVSR “AV (TM-CTC)”, LSTM-Transformer based AVSR “AV(LSTM/TM-CTC)”, and feature fusion-based AVSR “AV (feature fusion)”. In the experiments, the training set contains the speakers of the test set. Experimental results are shown in Table 5.

As shown in the experiment, if the training set includes all test set’s speakers regardless of their differences, CER will increase as the sentences in the dataset decrease in single modeling based on visual or audio data. For a single modality, the CER of video modeling is 5.2% to 8.0% lower than that of audio modeling. The CER of AVSR is lower by 16.9% to 17.1% than that of the single modality. Among different approaches of AVSR, the LSTM-Transformer approach put forward by the paper enjoys the best performance, achieving a minimum CER of 46.9% in recognizing low-resource language. This CER is 5.6% to 14.5% lower than those of the other two approaches. The Transformer-based fusion model takes the runner-up place in this comparison. With scarce data, feature fusion approaches do not see their loss decrease and the performance is worse than that of video modeling.

As the sampling of audio information is undertaken in a natural environment, noises are inevitable and interfere with audio models. However, video models, focusing on visual information, recognize speech contents with speakers’ lip movement. In this way, noises’ influence is reduced and the comparative experimental results of a multi-modality and single modality verify this reduction. In addition, the proposed AVSR model can significantly improve the recognition accuracy compared with the single modality.

Figure 5 shows the AVSR performance in recognizing initials. The results show that compared with audio recognition alone, the AVSR (with visual information) enjoys higher accuracy. The comparison results of (a) and (b) show that the chances are less for affricates to be confused with fricatives, which may be due to the fact that the lip shape of affricates differs greatly from that of fricatives; combining visual and audio information helps to discriminate these two kinds of phonemes. Similarly, the accuracy of recognizing nasals m, n, and ŋ also increases and the nasals are less likely to be confused with fricatives and semivowels w and j in (c) and (d). It may be because the lip movement of nasals is continuous but minor and that of fricatives and semivowels is obvious.

In addition, from the comparison of (e) and (f), it can be seen that the results of the two methods are similar, and the AVSR cannot significantly improve the confusion of aspirated stops, nasals, laterals, and voiceless fricatives. The reason may be that the pronunciations of these phonemes are quite different and can be distinguished by sounds. However, some of their lip movements have changed significantly, and some have no obvious changes. The former can increase the recognition accuracy, while the latter increases the confusion of some pronunciations, resulting in insignificant overall changes.

However, unaspirated stop p is more likely to be confused with aspirated stop p^h^, unaspirated fricative t with aspirated fricative t^h^, and unaspirated stop k with aspirated stop k^h^. These three pairs are bilabials, blade-alveolae, and velars, respectively. The unaspirated and aspirated phoneme differ slightly in the lip’s shape and mainly in the tongue’s position. It is difficult to tell the difference in the video mode but easier in the audio mode. Introducing video information, thus, hampers the model’s decision. Similarly, in affricates, unaspirated tɕ and aspirated tɕ^h^ are more likely to be confused, so do unaspirated ts and aspirated ts^h^.

Figure 6 shows AVSR’s performance in recognizing finals where the accuracy improves substantially. Compared with in audio modeling alone, diphthongs are less likely to be confused with monophthongs in AVSR modeling. It may be due to some diphthongs differing greatly from monophthongs in lip movement. The introduction of visual information, thus, greatly improves accuracy. In addition, nasal vowels are much less likely to be confused with compound vowels. Compared with initials, finals differ mainly in mouth shape and lip movement and slightly in tongue position, so the influence of visual information is more obvious.

### 4.3. Results of Speaker-Independent Experiments

To explore AVSR’s generalization onto different speakers, we also design speaker-independent experiments. In the experiments, the training set contains the speakers of the test set. The results are shown in Table 6.

With the speaker difference taken into consideration, a decrease in data leads to an increase in CER. Video-only modeling has a better performance than audio-only modeling. Fusion modeling has a better performance than single modeling with the CER dropping by 6.8% to 18.8%. The performance gap here is larger as compared with the modeling that does not take into account speaker difference. For fusion modeling approaches, LSTM-Transformer modeling still has the best performance with modeling using Transformer as both the encoder and decoder taking the second place. Training is impossible for unseen speakers in feature fusion modeling.

Figure 7 is a comparison of the experimental results of the single modality and multi-modality on overlapped speakers and unseen speakers.

Figure 7 shows the character error rate distributions and differences in different methods under different data sizes. We choose 9:1 as the training-test ratio in the experiment; 1800 sentences of audiovisual data are used, which contain the information of 4 speakers; 1620 sentences are used for training; 180 sentences are used for testing. The speaker-dependence experiments consist of training and testing the two models. The first training obtained model uses the speech of 4 speakers as the training set, and the test results are shown in Table 5; the second training obtained model uses only the speech of 3 speakers as the training set and then uses the speech of the fourth speaker to test the model. The test results are shown in Table 6.

Comparing the results of overlapped and unseen speakers as shown in Figure 7, we find that the CER of single and fusion modeling is higher when including speaker difference (the training set does not include test set’s speakers) than when excluding such a difference. However, the CER of fusion modeling is evidently lower than that of single modeling in both situations. The fusion modeling overall achieves better performance than single modeling. The LSTM-Transformer modeling we propose has the lowest CER, 17.2% lower than that of single modeling. LSTM-Transformer modeling is subject to a higher error rate when the data amount changes as compared with the modeling with Transformer as both the encoder and decoder and video-only modeling. However, the fluctuation is within a smaller margin, only 1.25% higher than the optimal result. Therefore, fusion modeling is more suitable at recognizing Tujia speeches with lower speaker dependence and fewer speakers.

## 5. Discussion

Because Tujia language has few native speakers, it is a typical low-resource language, so the collection of the corpus used in this paper is very difficult. The corpus used in this paper was collected by linguists in the Tujia language area. It has prominent problems such as high noise, small number of speakers, and variable collection angles, which make traditional ASR methods challenging to identify them. This paper designs an audiovisual fusion method for automatic recognition of Tujia language, which not only effectively solves the model’s dependence on Tujia speakers, but also achieves high recognition accuracy under the condition of few training samples. In particular, the end-to-end audiovisual fusion recognition system designed in this paper can supplement the visual modal information on the basis of the acoustic information, and finally obtain higher recognition performance.

In this paper, several experiments are designed to evaluate the performance of the AVSR model, which first proves that the end-to-end sequence-based modeling approach outperforms the state-transfer-based modeling approach due to the unique pronunciation rules of Tujia language. The AVSR model proposed in this paper can greatly improve the accuracy, and the confusion of phonemes is generally better than that of the single-modal model. Specifically, AVSR is significantly better than acoustic models in the recognition of affricates and fricatives, nasals and fricatives, and semivowels w and j, but is lower in aspirated and unaspirated stops; otherwise, the confusion between stops and nasals, lateral sounds, and unvoiced fricatives is similar to acoustic models, which indicates that AVSR is more effective in recognizing phonemes with obvious differences between pronunciation and lip movements, and has a poorer effect on phonemes with no obvious differences in pronunciation and lip movements. If the difference between the two influencing factors is opposite, the AVSR effect and the acoustic model are relatively close.

The proposed LSTM-Transformer audiovisual fusion model also significantly improves the generalization of the speakers. The comparison of the results of speaker-independent experiments shows that the difference in speakers will affect the accuracy. This is due to differences in speakers’ frequency and timbre leading to differences in extracted features of the same word. Differences also exist in lips’ deformation when different speakers say the same word. The method proposed in this paper can effectively improve the impact on the model performance caused by the above problems. Although it is greatly affected by the amount of data compared with other methods, the low-resource language data are generally less, and the impact on AVSR is limited. The proposed method can still work effectively on small datasets.

In conclusion, this method achieves higher accuracy on low-resource languages compared to traditional methods. In addition, the results of experiments on different modalities show that both audio-only models and video-only models have a certain dependence on the amount of data, and the character error rate will increase with the reduction in data, but compared with single-modal models, AVSR is more sensitive to changes in the amount of data. At the same time, feature fusion has more information redundancy, and the redundant information will affect the performance of the model and make the model no longer work.

Despite the above achievements, our work still has some limitations. In the process of comprehensive judgment of multi-modal models, the modal information with good performance will be affected by the modal with poor performance. Imbalance will lead to a decrease in the accuracy of the results, so in the next work, we will explore the effective use of multi-modal information to maximize the use of information and mutual assistance.

## 6. Conclusions

To protect the low-resource language (Tujia language), we apply ASR to semantic labeling of low-resource speeches. In order to solve the problem of poor performance of traditional ASR systems in low-resource language speech recognition, the audiovisual fusion modeling proposed in this paper effectively increases the accuracy. In addition, we put forward an AVSR modeling based on LSTM-Transformer. Under identical experimental conditions, the LSTM-Transformer model has a better accuracy than other models. In the LSTM-Transformer model, initials and finals are much less likely to be confused, with the recognition of finals achieving much higher accuracy. Finally, the generalization of the model to speakers is also significantly improved, which effectively solves the problem of excessive speaker dependence of the model caused by the lack of speakers in low-resource languages. In the future, we will further improve the audiovisual fusion strategy, in order to maximize the use of different modal information, thereby improving accuracy.

## Figures and Tables

**Figure 1 sensors-23-02071-f001:**
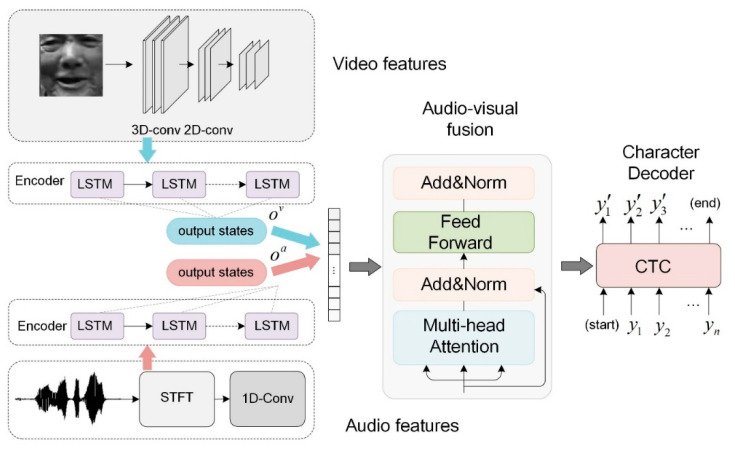
Illustration of Audiovisual Fusion.

**Figure 2 sensors-23-02071-f002:**
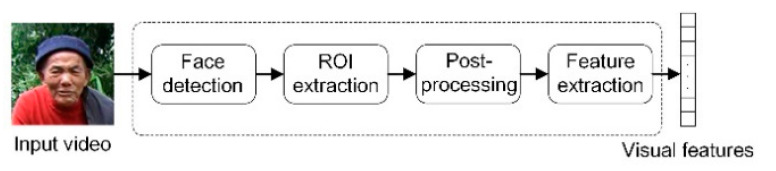
Extraction of visual features.

**Figure 3 sensors-23-02071-f003:**
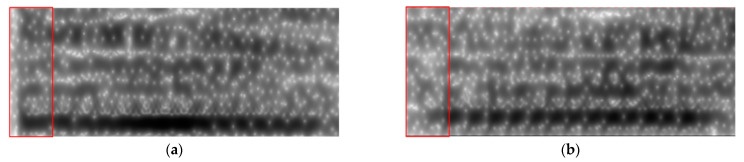
Spectrograms of different speakers pronouncing ai.

**Figure 4 sensors-23-02071-f004:**
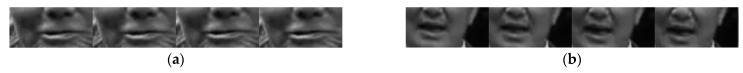
Lips’ shape of different speakers pronouncing *ai*.

**Figure 5 sensors-23-02071-f005:**
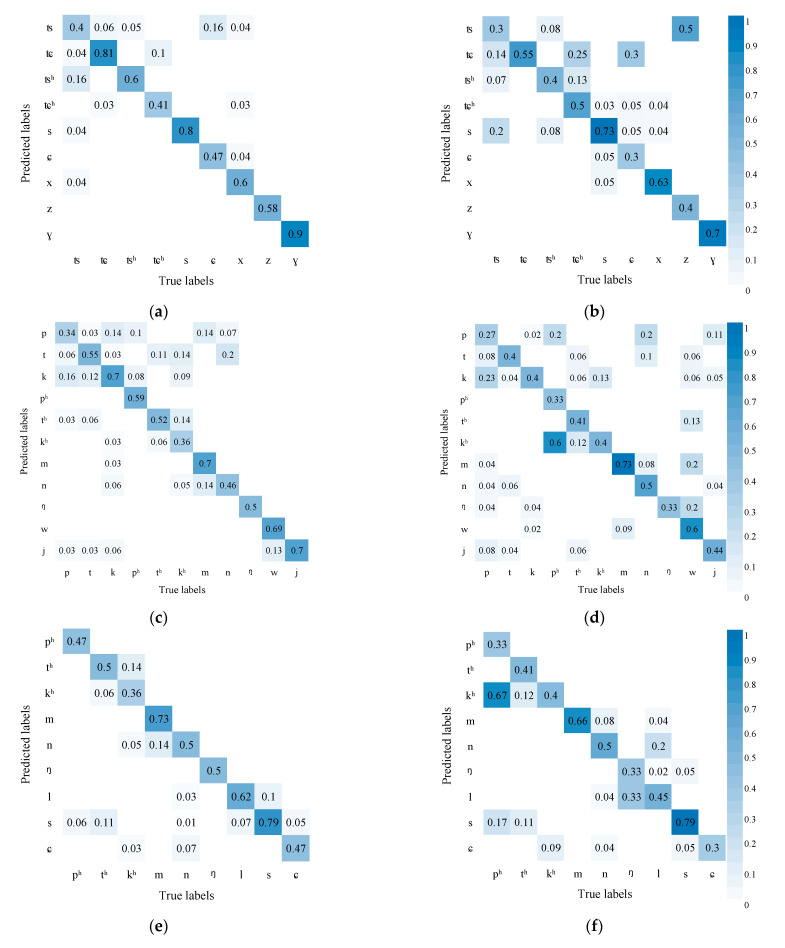
AVSR performance in recognizing initials (Left is the recognition results of AVSR, right is the recognition results of ASR): (**a**) Affricates and fricatives confusion matrix of AVSR; (**b**) Affricates and fricatives confusion matrix of ASR; (**c**) Fricatives, nasals, and semivowels confusion matrix of AVSR; (**d**) Fricatives, nasals, and semivowels confusion matrix of ASR; (**e**) Aspirated stops, nasals, laterals, and voiceless fricatives confusion matrix of AVSR; (**f**) Aspirated stops, nasals, laterals, and voiceless fricatives confusion matrix of ASR; (**g**) Aspirated and unaspirated stops and fricatives confusion matrix of AVSR; (**h**) Aspirated and unaspirated stops and fricatives confusion matrix of ASR.

**Figure 6 sensors-23-02071-f006:**
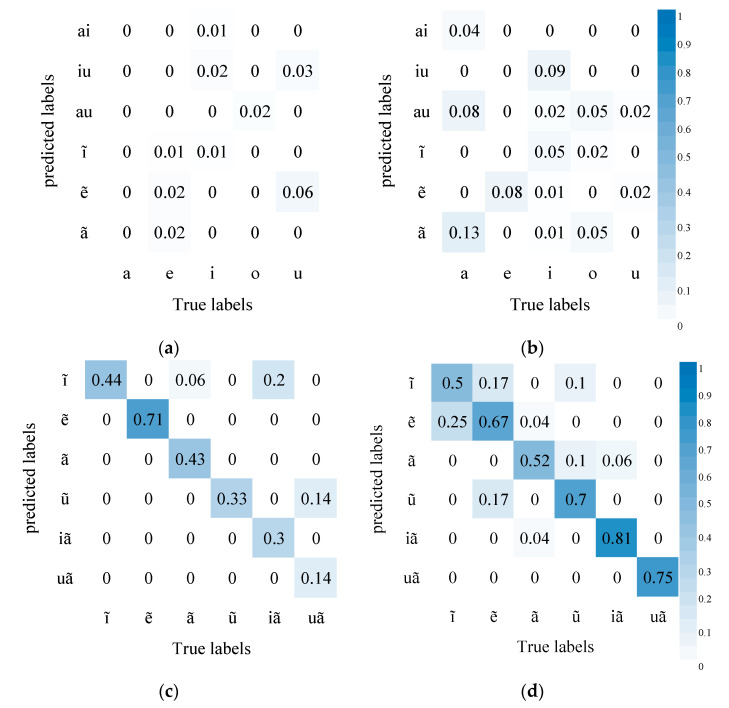
AVSR performance in recognizing finals: (**a**) Simple Finals and other finals Confusion matrix of AVSR; (**b**) Simple Finals and other finals Confusion matrix of ASR; (**c**) nasal vowels Confusion matrix of AVSR; (**d**) nasal vowels Confusion matrix of ASR.

**Figure 7 sensors-23-02071-f007:**
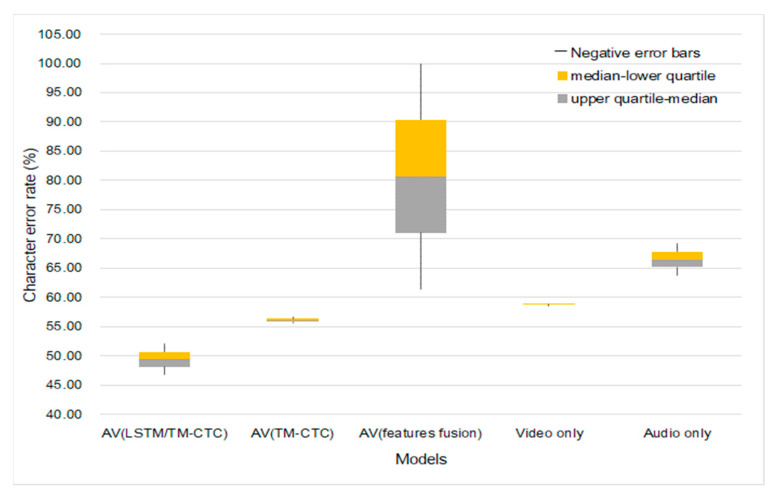
Models’ generalization onto different speakers.

**Table 1 sensors-23-02071-t001:** Datasets of Tujia language.

Name	Dates	Speakers	Sentences	Duration
Tima Legend	20 June 2017	Lu Bangzhu	349	12 m 59 s
Origins of Lu Family	19 June 2017		186	7 m 53 s
Festival Customs of Tujia People	17 June 2017		329	12 m 5 s
Marital Customs of Tujia People	17 June 2017		228	8 m 27 s
Construction of Stilted Buildings	17 June 2017		348	12 m
Experience of Being an Official	17 June 2017	Lu Chengcheng	155	6 m 23 s
Life of Lu Longwen	15 June 2017	Lu Longwen	122	3 m 55 s
Life of Lu Kaibai	14 June 2017	Lu Kaibai	43	2 m 23 s

**Table 2 sensors-23-02071-t002:** Initials of Tujia language.

			Bilabial	Supra-Dental	Blade-Alveola	Frontal	Velar
Stops	voiceless	unaspirated	p		t		k
		aspirated	p^h^		t^h^		k^h^
Affricates	voiceless	unaspirated		ts		tɕ	
		aspirated		ts^h^		tɕ^h^	
Nasals			m		n		ŋ
Laterals					l		
Fricatives	voiceless			s		ɕ	x
	voiced			z			ɣ
Semivowels			w			j	

**Table 3 sensors-23-02071-t003:** Finals of Tujia language.

Categorization	Finals
Simple Finals	i、e、a、ɨ、o、u
Compound Finals	ie、ei、uei、ai、uai、ia、ua、iu、ou、iau、au
Nasal Finals	ĩ、ẽ、ã、ũ、uẽ、iã、uã、iũ

**Table 4 sensors-23-02071-t004:** CER of different acoustic models recognizing different quantities of sentences.

Model	Quantity of Sentences				
	2000	3000	4000	5000	6000
HMM	91.03	79.27	76.28	73.19	69.97
GMM/HMM	90.02	82.42	77.41	73.01	67.74
GMM/HMM + LDA	91.10	82.43	77.62	72.86	67.52
GMM/HMM + LDA + MLLT	93.64	85.63	81.86	75.13	68.62
GMM/HMM + LDA + MLLT + SAT	97.67	83.60	79.83	76.66	69.32
HMM/LSTM	-	-	-	-	73.94
Transformer/CTC	66.0	61.7	58.6	53.3	48.2

**Table 5 sensors-23-02071-t005:** CER for different quantities of sentences in video-only, audio-only, and different fusion models.

Model	Quantity of Sentences		
	1000	1400	1800
Video-only	60.2%	59.2%	58.7%
Audio-only	68.0%	67.2%	63.8%
AV(TM-CTC)	60.1%	55.8%	55.6%
AV(LSTM/TM-CTC)	50.9%	50.2%	46.9%
AV (feature fusion)	-	61.7%	61.4%

**Table 6 sensors-23-02071-t006:** With the training set’s speakers excluded from the test set, the CER of audio, video, and different approaches of fusion modeling to recognize different quantity of sentences.

Model	Quantity of Sentences		
	1000	1400	1800
Video-only	61.4%	60.9%	58.8%
Audio-only	70.3%	69.7%	69.2%
AV(TM-CTC)	59.0%	58.1%	56.6%
AV(LSTM/TM-CTC)	63.7%	56.0%	52.0%
AV (feature fusion)	-	-	-

## Data Availability

Not applicable.

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
