# Peer review of "Improvement of Acoustic Models Fused with Lip Visual Information for Low-Resource Speech"

_sensors, 2023, doi:10.3390/s23042071_

Round 1

Reviewer 1 Report

The topic is interesting but the methods are technically not sound.

For a start, 3 speakers in train set and 1 speaker in test set is clearly not sufficient to draw any meaningful/valid conclusions from the presented results.

While it can be appreciated that native Tujia language speakers are not many, the work presented in this paper does not provide any conclusive evidence on any of the drawn conclusions. This is a major weakness of this paper.

Lines 61-63, state that Chinese corpus is used, but no evidence of using such a corpus can be seen in the results/discussion sections.

Line 99: please provide evidence for this statement by citing appropriate source or results.

Lines 135-141: How is the synchronization between audio and video modalities ensured? Clearly, synchronization is required.

Line 149: please correct spelling of corelated to correlated.

Lines 155-157: Why is the 3D -2D downsampling required? What is the "required dimension" that is being referred to in line 156? How has this "required dimension" been determined?

Line 159: please provide evidence that Tujia speeches are non-stationary....cite appropriate source

Line 161-162: what is the window length used in this study?

Line 163: what is meant by "signals are smooth within the time window".... smooth in what sense?

Line 163: "Supposing that signals are smooth within the time window...." Please justify the validity of this supposition.

Lines 182-183: How is the proposed multimodal fusion able to provide synchronous information of audio and video modalities and does not require forced alignment of the two modalities? please include evidence for this statement.

Section 3.1: The dataset used in this work is confusing. In line 259, it is stated that the dataset has 10348 short sentences. But the video data has 2105 short videos (line 270). It seems there is no video recording for all the uttered sentences. Combine this with the fact that only 4 speakers are used in this study, the dataset used in this study is improper and insufficient.

Since the dataset is inappropriate and insufficient, the results may not be accurate and hence the drawn conclusion seem invalid.

Lines 328-329: It is first stated that train-test ratio is 9:1 and then states that 3 speakers are used for training and 1 for testing. please clarify how the 9:1 ratio arrived at?

Line 336: How is the monophone extended to triphone model? what is the procedure involved?

Lines 341-343: for 2000, 5000 and 6000 sentences the CER is lower for triphone than monophone. But for 3000 and 4000 sentences, the opposite is true. what is the reason for this? the reason given on line 343-345 is not convincing.

In table 5, results of CER are for for 1000, 1400, 1800 where as in table 4 the CER values are for 2000, 3000, 4000, 5000, 6000 sentences. Why this variations. It would make sense to compare with same number of sentences.

While the paper is well presented in most parts, some spell checks are required. 

The experimental methodology is poor, the data set is neither appropriate nor sufficient. Hence, as a reviewer/reader, the results are not convincing. Therefore, the deduced conclusions may not even be valid. 

The paper requires extensive modification.

Reviewer 2 Report

This paper presents some improvements of the acoustic models fused with lip visual information for low-resource speech.
The specific gap addressed by this paper is the protection of endangered languages, as these have low resource characteristics.
The improvements this research proposes are based on an approach of audio-visual speech recognition (AVSR) based on LSTM-Transformer.

The state-of-the-art part is specifically well documented. The used evaluation metrics are correct and suitable for the experiments and the
references are appropriate.
The results are correctly described and the conclusions are properly drawn.

Round 2

Reviewer 1 Report

The authors have responded to the review comments satisfactorily by responding to the comments. 

However not all the responses are reflecting in the revised manuscript. It is recommended that the details given in your response be included in the revised/final manuscript. 

For example, the explanation given in response #13, #15, #18, needs to be included in the manuscript, to avoid ambiguity for readers.
